# *Physalis angulata* Calyces Modulate Macrophage Polarization and Alleviate Chemically Induced Intestinal Inflammation in Mice

**DOI:** 10.3390/biomedicines8020024

**Published:** 2020-02-05

**Authors:** David Rivera, Yanet Ocampo, Luis A. Franco

**Affiliations:** Biological Evaluation of Promising Substances Group, Department of Pharmaceutical Sciences, University of Cartagena, Carrera 50 No. 29-11, Cartagena 130014, Colombia; driveram@unicartagena.edu.co (D.R.); yocampob@unicartagena.edu.co (Y.O.)

**Keywords:** *Physalis angulata*, macrophages, inflammatory bowel disease

## Abstract

As part of the search for new bioactive plants from the Colombian Caribbean, the dichloromethane fraction of the calyces of *Physalis angulata* L. (PADF) was selected for its anti-inflammatory activity. In this work, we investigated the immunomodulatory effect of PADF in activated macrophages and during dextran sulfate sodium (DSS)-induced colitis. PADF displayed a low content of withanolides or phenolic compounds, and a higher content of sucrose esters, representative anti-inflammatory metabolites of the *Physalis* genus. The PADF fraction at 12.5 μg/mL prevented the induction of interleukin (IL)-1β, tumor necrosis factor (TNF-α), IL-6, IL-12, cyclooxygenase-2 (COX-2), and inducible nitric oxide synthase (iNOS) by lipopolysaccharide (LPS), while increased the levels of arginase (ARG1), IL-10, and mannose receptor C (MRC1). The polarization towards an anti-inflammatory profile was also observed in resting macrophages, without promoting the typical gene profile induced by IL-4, suggesting that PADF promotes a shift to a regulatory status rather than to an alternative one. In vivo, the administration of PADF to mice with chronic DSS-colitis reduced disease signs (i.e., body weight loss and colon shortening), and improved the histology score by diminishing the levels of pro-inflammatory cytokines and increasing the production of IL-10. Overall, results suggest that the regulatory effect on PADF towards macrophages might contribute to the therapeutic activity observed in the murine model of inflammatory bowel disease.

## 1. Introduction

The discovery of novel bioactive compounds from medicinal plants and animals continues to be an interesting/attractive strategy for the development of new drugs with a unique chemical and pharmacological profile; in fact, a new golden era for natural product research is anticipated from the recent technological advances and the current attention to naturally-derived drugs motivated by the 2015 Nobel Prize in Physiology or Medicine awarded to William C. Campbell, Satoshi Omura, and Youyou Tu for the discovery of avermectins and artemisinin, respectively [1].

Certainly, herbal medicines and metabolites have a high potential to treat a wide spectrum of diseases, and are used not only in developing countries, where up to 80% of the population still depends on traditional medicine and medicinal herbs for primary health care, but also in developed countries, where the influence of natural products in the pharmaceutical industry is remarkable [2,3]. Colombia, having one of the largest and complex biodiversity, has enormous potential to generate scientific knowledge to establish an innovative, sustainable, and profitable local industry for natural health products. The *Physalis* genus (Solanaceae), which includes 90 species native to America, constitutes an excellent exemplification of the exploitation of endemic plants to develop valuable commercial products. Recently, increasing attention on the phytochemistry and pharmacological effect of *Physalis* has led to the development of commercial formulations, varying from functional foods and cosmetics to health products [4]. This has been accomplished with species cultivated on a large scale (i.e., *Physalis peruviana* and *Physalis alkekengi*) [5,6], but progress is limited with those growing wildly or cultivated at small scale. Our ongoing bioprospecting investigation on the Colombian Caribbean led to the identification of the anti-inflammatory potential of the calyces of *Physalis angulata* L. [7]; a wild species with extensive cultivation potential in tropical areas [8]. Although *P. angulata* is already recognized as a model of wide interest for its pharmacological and nutritional properties [9], few studies have focused on the calyces, a characteristic organ of the *Physalis* genus that is developed after pollination by modified leaves, forming a papery lantern-like structure that covers and protects the fruit [10]. Given their physiological role, calyces are supposed to constitute a prolific reservoir of bioactive metabolites, in comparison with the rest of the plant. Focusing on the anti-inflammatory potential of *P. angulata*, numerous studies have demonstrated strong immunomodulatory function of several derivatives, with recent reports revealing the immunosuppressive effects of some withanolides due to regulation of macrophage polarization [11].

This work describes a pharmacological study of the dichloromethane fraction from *P. angulata* calyces, including the assessment of its immunomodulatory effect in activated macrophages, and mice with chemically induced intestinal inflammation.

## 2. Experimental Section

### 2.1. Reagents and Chemicals

Solvents, Dulbecco’s Modified Eagle Medium (DMEM), Roswell Park Memorial Institute medium (RPMI), antibiotics (penicillin/streptomycin), lipopolysaccharide (LPS) from *Escherichia coli* 0127:B8, *O*-dianisidine, and hematoxylin-eosin (H&E) were purchased from Sigma-Aldrich (St. Louis, MO, USA). Fetal Bovine Serum (FBS) was acquired from Gibco (Sao Paulo, Brazil). Recombinant mouse interleukin (IL)-4 was obtained from MACS/Miltenyi Biotec (Bergisch Gladbach, Germany). Dextran sulfate sodium (DSS; MW 36–50 kDa) was purchased from MP Biomedicals (Santa Ana, California, USA). Hexadecyltrimethylammonium (HTAB) and hydrogen peroxide (H_2_O_2_) were obtained from Calbiochem^®^ (San Diego, CA, USA).

### 2.2. Plant Collection and Preparation of the Dichloromethane Fraction from P. angulata

The calyces of *Physalis angulata* L were collected at Pueblo Nuevo, Bolívar, Colombia (10°44’ N; 75°15’ W) and identified by Felipe Cardona Naranjo in the herbarium of the University of Antioquia (Medellin, Colombia), where a voucher specimen was deposited (HUA175328). Dried calyces were employed to obtain the dichloromethane fraction (PADF) as previously reported by Rivera et al. [7] and fully described in the Appendix A.

### 2.3. Cell Culture

The RAW 264.7 macrophage cell line (TIB-71™) was obtained from the American Type Culture Collection (ATCC; Manassas, VA, USA) and maintained in DMEM with 10% FBS at 37 °C and 5% CO_2_.

### 2.4. Macrophage Activation and Gene Expression

RAW 264.7 macrophages (750,000 cells/well) were seeded in 6-well plates for 48 h, then treated with PADF (0–12.5 μg/mL) for 1 h, and stimulated with LPS (1 μg/mL) or IL-4 (40 ng/mL) to induce an inflammatory M(LPS) or an alternative M(IL4) profile, respectively. Six hours after stimulation, RNA was extracted using the RNeasy Mini Kit from Qiagen (Valencia, CA, USA) and the cDNA was synthetized with the Transcriptor Universal cDNA Master (Roche, Mannheim, Germany). Subsequently, the cDNA was used to assess changes in gene expression (IL-1β, tumor necrosis factor (TNF-α), IL-6, IL-12, cyclooxygenase-2 (COX-2), inducible nitric oxide synthase (iNOS), arginase (ARG1), IL-10, transforming growth factor, beta 1 (TGF-β1), and mannose receptor C (MRC1) using a LightCycler 96 and the Fast Start Essential DNA Green Master (Roche, Mannheim, Germany). Specific primers (Eurofins Genomics, Huntsville, USA) sequences are listed in Table 1. Gene expression was normalized to β-actin, using duplicate cycle threshold (CT) values analyzed by the comparative CT (ΔΔCT) method.

### 2.5. Animals

Four-week-old female CD-1(ICR) mice were obtained from the Instituto Nacional de Salud (Bogotá, Colombia). Animals were housed in filtered-capped polycarbonate cages and kept in a controlled environment (22 ± 3 °C, 65–75% humidity, under a 12 h light/darkness cycle) with access to food and water ad libitum. All the experiments were designed and conducted in accordance with local and international regulations (EU Directive 2010/63/EU) and approved by the Committee of Ethics in Research of the University of Cartagena (Project Approval No 81 from 13 August 2015).

### 2.6. DSS-Induced Colitis Model

Animals were randomized into control (vehicle), colitis-control (DSS group), and treatment groups (*n* = 6–17). Colitis was induced with 4% (*w*/*v*) DSS in drinking water during two seven-day cycles, separated by ten days of regular water. Two doses of PADF (5 and 10 mg/Kg) were administered intraperitoneally (i.p.) during seven days after DSS withdrawal, while the control group was treated with saline. Body weight and disease activity index (DAI) were monitored exhaustively [12]. Macroscopic parameters, such as colon length and weight/length ratio were also measured after mice sacrifice.

### 2.7. Histologic Analysis

Colonic tissue samples were preserved in buffered formalin and embedded in paraffin, 5 μm sections were cut and stained with H&E. Damage to the epithelial structure and cell infiltration in the colonic tissue were qualified using a 0 to 4 score (0: none, 1: mild, 2: moderate, 3: high and 4: very high) by a blind pathologist using light microscopy (Axio Lab A1, Zeiss, Oberkochen, Germany).

### 2.8. Myeloperoxidase (MPO) Activity Assessment

MPO activity was measured according to the method described by Bradley et al. [13]. Colonic tissue samples were homogenized using a potassium phosphate buffer containing 0.5% HTAB, subjected to three freezing/thawing cycles in liquid N2, and finally centrifuged to obtain a supernatant used to determine MPO levels with *O*-dianisidine, HTAB, and H_2_O_2_. MPO activity was expressed as enzyme activity units per mg of protein. Protein concentration was measured using the Bradford assay (Bio-Rad, Laboratories Inc., Hercules, CA, USA).

### 2.9. Determination of Pro-Inflammatory Cytokines in Colon and Mesenteric Lymph Nodes

On one hand, colonic tissue was subjected to mechanical homogenization and centrifugation in Tissue Protein Extraction Reagent (T-PER, Thermo Fisher Scientific, Waltham, Massachusetts, USA), supplemented with complete protease inhibitor cocktail (Roche). The levels of IL-1β, IL-6, IL-10, and TNF-α were determined by ELISA (Thermo Fisher Scientific, Waltham, MA, USA) and normalized to total protein, as measured by Bradford assay (Bio-Rad). On the other hand, mesenteric lymph nodes (MLN) from healthy mice (*n* = 3) were used to obtain a single cell suspension that was cultured in 24-well plates (1 × 10^6^ cells/well) using RPMI-1640 medium, supplemented with 10% FBS and antibiotics, at 37 °C and 5% CO2. After 1 h, the cells were treated with PADF (12.5 μg/mL) for an additional hour and then stimulated with LPS (1 μg/mL) for 24 h. Finally, the supernatants were collected to determine IL-6 and MCP-1 levels by ELISA (Thermo Fisher Scientific).

### 2.10. Statistical Analysis

Results are expressed as the mean ± standard error of the mean (SEM) of at least two independent experiments. Data were analyzed using one-way analysis of variance (ANOVA), followed by Dunnett’s post hoc test. Values of *p* < 0.05 were considered significant.

## 3. Results

### 3.1. Extraction of P. angulata and Preliminary Phytochemical Study of PADF

A total of 6.32 g of the dichloromethane fraction (PADF) were obtained from 234 g of dried calyces (2.70% yield) from *P. angulata*. The phytochemical characterization of F0-2 was performed by HPLC-DAD to generate a reliable chemical fingerprint to control the quality of future preparations and to predict bioactivity, as described in the Appendix A. However, the identity of the characteristic peaks of F0-2 remains unknown (Appendix A), since there is a notorious lack of commercial standard compounds for *Physalis* genus [14].

Moreover, considering that sucrose esters are described as the main anti-inflammatory metabolites from the calyces of *Physalis* [15] and several anti-inflammatories withanolides have been isolated from *P. angulata*, we quantified the total content of sucrose esters [16] and withanolides [17] (Appendix A). The results revealed that calyces from *P. angulata* are an important source of sucrose esters whereas the withanolides were at a low level (Appendix A).

### 3.2. PADF Modifies the Gene Expression Profile of Resting and LPS-Stimulated RAW 264.7 Macrophages Promoting an Anti-Inflammatory Polarization

To gain insight into the effects of PADF on macrophages differentiation, RAW 264.7 cells were treated with the test fraction (6.25 and 12.5 μg/mL) prior to polarization towards two contrasting activation states: inflammatory/LPS-activated [M(LPS)] and alternative/IL4-activated [M(IL4)]. As expected, inflammatory M(LPS) macrophages expressed higher mRNA levels of IL-1β, tumor necrosis factor (TNF)-α, IL-6, IL-12, cyclooxygenase (COX)-2, and inducible nitric oxide synthase (iNOS) (pro-inflammatory genes), as well as a strong reduction of arginase (ARG)1, IL-10, and mannose receptor C (MRC1) (anti-inflammatory genes) in comparison to the resting control (M0) (Figure 1A,B). In contrast, PADF significantly prevented the induction of pro-inflammatory genes, in a dose-dependent manner (Figure 1A). Additionally, the treatment with PADF significantly promoted the expression of anti-inflammatory genes (ARG1, IL-10, and MRC1) (Figure 1B).

Similarly, when evaluating these anti-inflammatory genes on M0 macrophages treated with PADF, a significant increase in the expression of ARG1, IL-10, TGF-β1, and MRC1 mRNA was also observed (Figure 1C); suggesting that PADF might be capable of inducing a macrophage polarization shift towards an anti-inflammatory profile. However, the well-known expression of ARG1 or MRC1 by alternatively activated macrophages M(IL4) was not boosted by test fraction; moreover, no effects on IL-10 were observed (Figure 1D).

Finally, the 24 h accumulation of nitrite (marker of iNOS activity), urea (marker of ARG1 activity), and IL-10 (immunosuppressive cytokine) were measured using the Griess reaction [18], enzymatic detection or ELISA as detailed in Appendix A. Results corroborated that PADF inhibited the production of nitrite by inflammatory M(LPS) macrophages, but did not increase the levels of urea in the case of alternative M(IL-4) macrophages (Appendix A). However, when studying IL-10, the test fraction significantly inhibited its production by M(LPS) but had no effect in M(IL-4).

### 3.3. P. angulata Fraction (PADF) Ameliorates DSS-Induced Colitis

DSS intake produced a significant body weight decrease on the colitis-control group by the end of the last DSS cycle, while the administration of PADF at 5 and 10 mg/Kg/day managed to reverse this disease sign with a full recovery of their body weight (0.95% and 0.62% weight gain, respectively) without reaching statistical significance. Moreover, disease activity index (DAI) score also tended to be diminished by PADF treatment (Figure 2A). A macroscopic evaluation was also performed in order to further assess the severity of the disease and the effect of the treatment, where colon length; a very common and important parameter in DSS-induced colitis, was measured. As can be seen in Figure 2B, both doses of PADF were able to significantly decrease the colon shortening when compared to the colitis-control group. The histologic analysis also showed that treatment with PADF (10 mg/Kg) produced a significant improvement in terms of edema, immune cell infiltration, and tissue structure integrity (Figure 2C) diminishing the histological score to 3.25 ± 0.33 in comparison to the colitis-control group (5.17 ± 0.33).

Besides histologic analysis, polymorphonuclear cell infiltration can be indirectly determined by measuring the levels of myeloperoxidase (MPO) in the tissue. As shown on Figure 3A, DSS intake significantly increased the MPO activity in the colon (91.60 ± 11.00 U/mg protein) compared to the control group (52.44 ± 9.38 U/mg protein). In accordance with the observations on the histologic slides, treatment with PADF (10 mg/Kg) was able to significantly reduce the MPO activity by 44% approximately. Similarly, the lower dose of PADF (5 mg/Kg) exerted a significant reduction of MPO activity, confirming indirectly that the test fraction reduced the infiltration of pro-inflammatory cells.

Furthermore, the administration of PADF was associated with a diminished expression of pro-inflammatory cytokines and over-expression of anti-inflammatory cytokines. Indeed, mice treated with PADF showed reduced levels of IL-1β and TNF-α, while levels of IL-10 were significantly increased. A non-significant decrease of IL-6 levels was also observed (Figure 3A).

Additionally, Figure 3B shows the effect of PADF on the production of IL-6 and MCP-1 by LPS-stimulated mesenteric lymph nodes (MLN) cells. The test fraction (12.5 μg/mL) did not affect the production of any of the soluble mediators on naïve cells. However, it significantly reduced the production of IL-6 and MCP-1 by LPS-stimulated MLN cells.

## 4. Discussion

This study reveals the pharmacological potential of the dichloromethane fraction (PADF) of *P. angulata* calyces showing its capacity to ameliorate established DSS-colitis in mice. Our results suggest that PADF exerts its anti-inflammatory effect by inducing macrophage polarization towards a regulatory profile. In fact, PADF promotes a strong reduction of pro-inflammatory genes expression and a significant induction of anti-inflammatory markers, including IL-10, after stimulation with LPS or in resting macrophages. Moreover, a significant reduction of pro-inflammatory cytokines and the induction of IL-10 were observed in the colonic tissue of PADF-treated mice.

*Physalis angulata* is an annual herb that is distributed from North America (southeastern United States) to Central and South America (to Paraguay) and the Antilles [19]. In Colombia, this species grows from 0 to 1800 m above sea level and could be found throughout the country, where it constitutes one of the most common representatives of the *Physalis* genus. While the fruits are consumed by native indigenous communities, the aerial parts (including calyces) and fruits are commonly employed for its anti-inflammatory and anti-microbial properties [20]. So far, at least a dozen studies have demonstrated the ability of extracts, fractions, and compounds from *P. angulata* to inhibit inflammation and modulate the response of immune cells, including macrophages. To our knowledge, macrophages are the most commonly employed immune cells to study *P. angulata* derivatives, with all the reports employing LPS-stimulated macrophages [M(LPS)] and the quantification of pro-inflammatory mediators (such as NO, IL-6, and TNF-α) to demonstrate their anti-inflammatory properties. However, with the exception of Rivera et al. [7], none of these reports focused on the calyces. Furthermore, no information is available regarding the impact of *P. angulata* on the expression of anti-inflammatory markers (i.e., IL-10 or TGF-β1) by M(LPS) macrophages or the modulation of resting macrophages (M0) or alternatively activated macrophages [M(IL-4)]. Our study is the first to demonstrate that treatment with PADF modulated macrophage polarization towards a non-inflammatory phenotype. Specifically, PADF shifted the inflammatory status of M(LPS) with a potent reduction of iNOS, COX-2, IL-1β, IL-6, and IL-12; that was accompanied by a significant increase of anti-inflammatory genes (ARG1, IL-10, MRC1, and TGF-β) which are poorly or not expressed by M0 or M(LPS) [21]. In all probability, this PADF-induced polarization does not promote the transition from the classically activated M(LPS) to the alternatively activated phenotype M(IL-4) since: (1) M0 treated with PADF not only expressed higher levels of alternative activation markers (i.e., ARG1 or MRC1), but also exhibited higher expression of anti-inflammatory genes (IL-10 and TGF-β); or (2) The treatment with PADF did not enhance the expression of alternative activation markers (i.e., ARG1 or MRC1) when stimulated with IL-4. In this case, results suggest that the phenotype promoted by PADF is similar to that induced by IL-10 [M(IL-10) or M2c] which is more related to the suppression of immune responses and tissue remodeling through the expression of markers, such as IL-10, TGF-β, and MRC1 [22].

As *P. angulata* is also employed in folk medicine to treat gastrointestinal discomfort [23], and considering the crucial role of macrophages in the pathogenesis and resolution of inflammatory bowel disease, we continued our study with the assessment of PADF using a therapeutic setting of DSS-induced colitis in mice. Results demonstrated that administration of PADF (10 mg/Kg/day) to mice with DSS chronic colitis significantly reversed the colon shortening, reduced the histological damage, neutrophil infiltration (as showed by MPO activity assay), and the production of inflammatory cytokines (IL-1β and TNF-α). In addition, PADF induced the production of IL-10 in DSS-treated mice, a key immunoregulatory cytokine for the maintenance of intestinal homeostasis. In mice, macrophages are considered one of the most important sources of IL-10 to maintain and minimize mucosal immunopathology by promoting Foxp3 expression in regulatory T cells (Treg) and their function during colitis [24]. Hence, it is plausible that PADF significantly ameliorates DSS-colitis by means of macrophage transition towards an anti-inflammatory phenotype. This is reinforced by the decreased production of IL-6 and MCP-1 by activated MLN cells, mediators that also play a pivotal role in the migration and attraction of inflammatory macrophages to colonic tissue when damage is induced by DSS [25]. Further research is needed to undoubtedly demonstrate whether the effect of *P. angulata* is mediated by intestinal macrophages instead of other cell populations.

While this is not the first study describing the intestinal anti-inflammatory activity of *P. angulata*, since recently Almeida Junior et al. [26] demonstrated that a CO_2_ standardized extract from the aerial parts of the plant—PACO2—effectively prevented the colonic damage induced by TNBS in rats; this is the first report of the bioactivity of calyces from *P. angulata* in a model of colitis. Moreover, our results revealed their potential to treat, instead of preventing, established intestinal inflammation.

Another important result of the present study was the confirmation of significant amounts of glycosides and withanolides; secondary metabolites that might be responsible for the bioactivity, as previously reported by Rivera et al. [7]. Up to date, all the studies related to the anti-inflammatory activity of *P. angulata* were directed to the assessment of withanolides such as physalins, aromaphysalins, physangulatins, and whitangulatins, labdane-type diterpenoids, such as physangulatosides, or phenol glycosides, namely physanguloside A. Here we found that sucrose esters tended to be the more abundant class of compounds in PADF; therefore, it is likely that the pharmacological effects of this fraction are related to their presence. In accordance, they have been described as the major anti-inflammatory compounds from calyces, fruits, or the sticky coat of several *Physalis* species [15,27,28].

Overall, we have identified PADF as a regulator of macrophage activation and inflammation. Our data suggest a potential therapeutic role of *P. angulata* in inflammatory bowel disease.

## Figures and Tables

**Figure 1 biomedicines-08-00024-f001:**
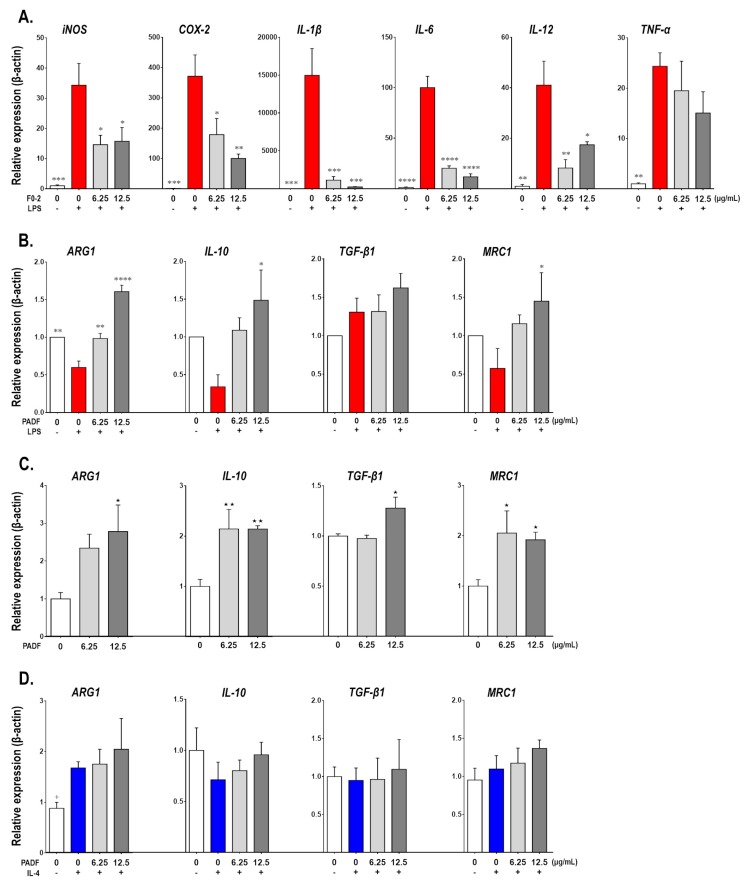
*Physalis angulata* fraction (PADF) polarizes macrophages towards an anti-inflammatory profile. Naïve RAW 264.7 macrophages were treated with PADF (6.25 and 12.50 μg/mL) or vehicle (positive and negative controls) and then stimulated with LPS [M(LPS)] or IL-4 [M(IL4)]. The expression of pro-inflammatory gene markers was quantified for inflammatory M(LPS) macrophages (**A**); while the expression of anti-inflammatory markers was assessed for inflammatory M(LPS) (**B**), naïve (**C**), and alternatively M(IL-4) (**D**) macrophages. All graphs show the mean ± SEM of at least three independent experiments (*n* = 4–9). * *p* < 0.05, ** *p* < 0.01, *** *p* < 0.001, **** *p* < 0.0001 significantly different from M(LPS) control; + *p* < 0.05 significantly different from M(IL-4) control; or * *p* < 0.05, ** *p* < 0.01 significantly different from naïve control (ANOVA, Dunnett’s multiple comparisons test).

**Figure 2 biomedicines-08-00024-f002:**
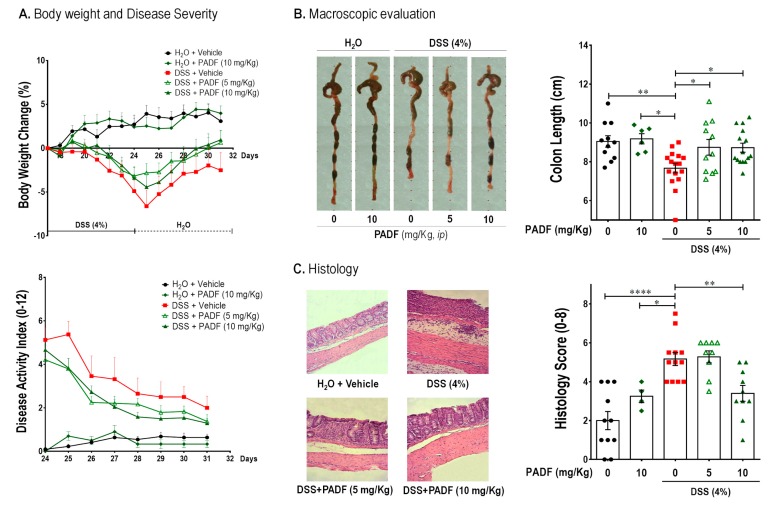
Chronic dextran sulfate sodium (DSS)-induced colitis is effectively treated by *P. angulata* fraction (PADF). Chronic colitis was established with two cycles of DSS (4%) interrupted by normal drinking water. Affected mice (*n* = 11–16 per group) were treated with PADF (5 and 10 mg/Kg/day, *ip*) or vehicle (saline) for 7 days. PADF alleviated colitis promoting a trend of increased body weight and reduced disease activity index (DAI) by the end of the experiment (**A**). Moreover, PADF-treated mice showed significantly longer colons (**B**) and lower histological scores with marked reduction of epithelial damage and cell infiltration (**C**). Pictures from representative animals are shown. Results represent the mean ± SEM (*n* = 6–18) of three independent experiments. * *p* < 0.05, ** *p* < 0.01, **** *p* < 0.0001 significantly different from DSS-colitis control, as calculated by ANOVA (Dunnett’s multiple comparisons test).

**Figure 3 biomedicines-08-00024-f003:**
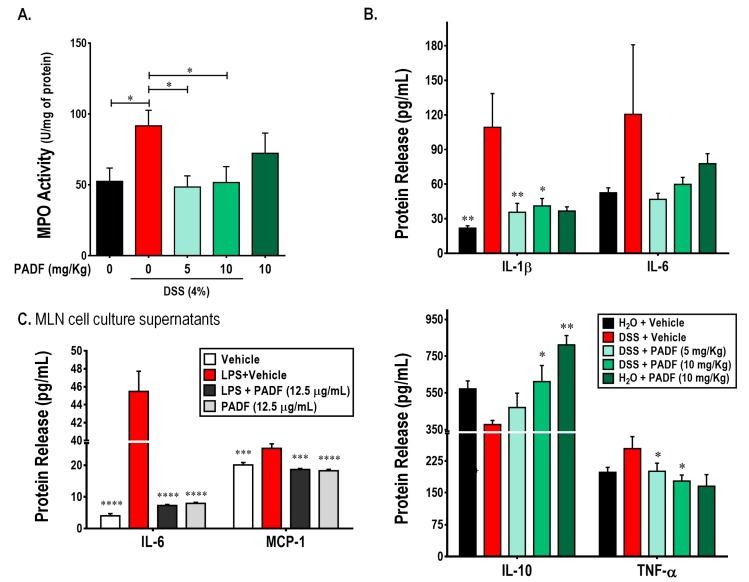
*P. angulata* fraction (PADF) exerts immunoregulatory effect in cells from colon and mesenteric lymph nodes (MLNs). The administration of PADF to mice with chronic colitis induced a significant reduction in the infiltration of neutrophils as demonstrated by a strong reduction of myeloperoxidase (MPO) activity (**A**). In addition, PADF regulated the levels of pro-inflammatory cytokines (IL-1β and TNF-α), while significantly induced the production of IL-10, an anti-inflammatory cytokine (**B**). PADF also reduced the release of IL-1β and IL-6 by MLN cells activated with LPS (**C**). Results represent the mean ± SEM from at least two independent experiments (*n* = 6–16). * *p* < 0.05, ** *p* < 0.01, *** *p* < 0.001, **** *p* < 0.0001 from DSS-colitis control group or LPS-activated MLN cells (ANOVA, Dunnett’s multiple comparisons test).

**Table 1 biomedicines-08-00024-t001:** Sequences of Primers used for Real-time PCR analysis.

Gene Symbol	Official Name	Primer Sequence ^1^
ACTB (β-actin)	Actin, beta	F: TAGGCGGACTGTTACTGAGC R: TGCTCCAACCAACTGCTGTC
ARG1	Arginase	F: TCGTGTACATTGGCTTGCGA R: GCCAATCCCCAGCTTGTCTA
PTGS2 (COX-2)	Prostaglandin endoperoxide synthase 2	F: TTCAACACACTCTATCACTGGC R: AGAAGCGTTTGCGGTACTCAT
IL-1β	Interleukin 1 beta	F: CTTCCTTGTGCAAGTGTCTG R: GCCTGAAGCTCTTGTTGATG
IL-6	Interleukin 6	F: CTGCAAGAGACTTCCATCCAG R: AGTGGTATAGACAGGTCTGTTGG
IL-10	Interleukin 10	F: TGCCTGCTCTTACTGACTGG R: CTGGGAAGTGGGTGCAGTTA
IL-12	Interleukin 12	F: TGGTTTGCCATCGTTTTGCTG R: ACAGGTGAGGTTCACTGTTTCT
iNOS	Nitric oxide synthase 2, inducible	F: ACATCGACCCGTCCACAGTAT R: CAGAGGGGTAGGCTTGTCTC
MRC1	Mannose receptor, C type 1	F: GCTTCCGTCACCCTGTATGC R: TCATCCGTGGTTCCATAGACC
TGF-β1	Transforming growth factor, beta 1	F: ACTGGAGTTGTACGGCAGTG R: TCATGTCATGGATGGTGCCC
TNF (TNF-α)	Tumor necrosis factor	F: ACCCTCACACTCAGATCATC R: GAGTAGACAAGGTACAACCC

^1^ F: Forward (5′→3′); R: Reverse (5′→3′).

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
