# Peer review of "Physalis angulata* Calyces Modulate Macrophage Polarization and Alleviate Chemically Induced Intestinal Inflammation in Mice"

_biomedicines, 2020, doi:10.3390/biomedicines8020024_

Round 1
Reviewer 1 Report
This is a well design study that was performed to assess anti-inflammatory activity of dichloromethane fraction of the calyces of Physalis angulata. Immunomodulatory effect of PADF was evaluated using cell culture model of activated macrophages and animal model of colitis. PADF displayed ability to prevent the LPS-mediated induction of IL-1β, TNF-α, IL-6, IL-12, COX-2, and iNOS while increased the levels of ARG1, IL-10, and MRC1. The polarization towards an anti-inflammatory profile was also observed in resting macrophages, without promoting the typical gene profile induced by IL-4. Overall, these results suggest that PADF promotes a shift to a regulatory status rather than to an alternative one. The data support the conclusions, however the choice of cell line needs more justification? Can this effect be observed using human immunocompetent cell from peripheral blood?
Author Response
Dear reviewer 1,
Thank you for taking time out of your busy schedule to review our manuscript. We appreciate your comments regarding the design of the study and think that your questions are appropriate. First, we employ the monocyte/macrophage cell line RAW 264.7 routinely in our lab since we are constantly testing dozens of plant-derived extracts/fractions/compounds, therefore these cells are a most suitable option in comparison to primary culture. As thoroughly described in the literature, this mouse-derived cell lineage is the most commonly used in biomedical research to study macrophage-mediated immune, metabolic and phagocytic function. Hence, since we had previous cytotoxicity and bioactivity data of the P. angulata fraction using these cells, we decided to use them for this specific project. We agree with the reviewer that using cells from human peripheral blood might have been interesting to reinforce our conclusions; however, in our opinion, they were not imperative and in consequence, they were not included in the project filed for authorization within the ethical committee of our university. Therefore, we are not able to perform experiments using samples from human origin in this short time. Anyway, experiments using cells from human tissue are warranted in our future work.
Reviewer 2 Report
In this study, the authors investigate the effect of Physalis angulate calyces on macrophage polarization and on a mouse model of intestinal inflammation.
The experiments are well described and nicely presented. However, the data presented in the paper are somehow disconnected, as there is no direct evidence that the effect of Physalis angulate calyces in the described in vivo model rely on their anti-inflammatory in vitro effects on macrophages. Moreover, some conclusions could be better corroborated by supplemental experiments.
MAJOR POINTS:
Fig. 1: While the anti-inflammatory effects of Physalis angulata calyces on LPS-treated macrophages are very clear, the data on resting and IL-4-treated macrophages are not solid. It would be interesting to see whether Physalis angulata calyces also affect protein levels, and not only mRNA, of some of the altered factors. This is particularly important for anti-inflammatory markers, as the urea levels shown in supplementary figures are not affected by Physalis angulata calyces, while ARG1 mRNA are induced. Moreover, IL-4 treatment does not show strong effects on typical anti-inflammatory genes, which is surprising. The authors should assay other markers (for example Fizz1) and/or try different IL-4 concentrations. As they are presented, it is difficult to conclude that Physalis angulata calyces and IL-4 do not synergize, as IL-4 is not playing any effect at all. Additionally, it would be interesting to see whether Physalis angulata calyces exert the same anti-inflammatory activity on other cellular models (other immune cell populations and intestinal epithelial cells.)
Fig. 2: It is not clear whether the data presented in the two graphs of panel 2A are statistically significant or not. The authors should clarify it in the manuscript.
Fig. 3: The authors show that Physalis angulata calyces in vivo administration reduces MPO activity in intestinal samples from DSS-treated animals. While this result correlates with the results presented in Fig. 2, it is not clear how to link them with results in Fig.1. Have the authors investigated whether Physalis angulata calyces affect neutrophil gene expression and inflammatory response? Similarly, experiments on MLN do not correlate with the data on macrophages, as MLN are composed by different immune populations.
MINOR POINTS:
Fig. 1: Physalis angulata calyces concentration in the figures is measured as mg/kg, while the legend states ug/ml.
Fig. 2: The authors should state the number of animals analyzed in fig. 2B and 2C, as the legend says “three independent experiments”.
Author Response
Dear reviewer 2,
Thank you for taking the time to review our manuscript. We read your comments and suggestions very carefully and modified the document to answer all your requests. We hope that these modifications meet your expectations. In addition, we prepared a point-by-point response to all your comments to provide further clarification.
Comment: The experiments are well described and nicely presented. However, the data presented in the paper are somehow disconnected, as there is no direct evidence that the effect of Physalis angulate calyces in the described in vivo model rely on their anti-inflammatory in vitro effects on macrophages.
Response: We understand your comment and revised the entire document to make sure that the entire document reflects that results obtained using macrophages in vitro are only a possible link to the inhibition of the intestinal inflammation induced by DSS in mice. It was never our intention to imply that there was a direct correlation between both scenarios (macrophages in vitro and DSS model); instead, we made an effort to clarify this within the discussion (“Hence, it is plausible that PADF significantly ameliorates DSS-colitis by means of macrophage transition towards an anti-inflammatory phenotype… Further research is needed to undoubtedly demonstrate whether the effect of P. angulata is mediated by intestinal macrophages instead of other cell populations”) and in the title of the paper where both effects were kept separated. To improve this distinction, we performed the following changes in the ABSTRACT:
The sentence “In accordance” was replaced for “In vivo”. The sentence “In conclusion, PADF is a regulator of macrophage activation with a potential therapeutic role in the context of inflammatory bowel disease” was replaced for “Overall, results suggest that the regulatory effect on PADF towards macrophages might contribute to the therapeutic activity observed in the murine model of inflammatory bowel disease”.Comment: Fig. 1: While the anti-inflammatory effects of Physalis angulata calyces on LPS treated macrophages are very clear, the data on resting and IL-4-treated macrophages are not solid. It would be interesting to see whether Physalis angulata calyces also affect protein levels, and not only mRNA, of some of the altered factors. This is particularly important for anti-inflammatory markers, as the urea levels shown in supplementary figures are not affected by Physalis angulata calyces, while ARG1 mRNA are induced. Moreover, IL-4 treatment does not show strong effects on typical anti-inflammatory genes, which is surprising. The authors should assay other markers (for example Fizz1) and/or try different IL-4 concentrations. As they are presented, it is difficult to conclude that Physalis angulata calyces and IL-4 do not synergize, as IL-4 is not playing any effect at all. Additionally, it would be interesting to see whether Physalis angulata calyces exert the same anti-inflammatory activity on other cellular models (other immune cell populations and intestinal epithelial cells.)
Response: We agree that the measurement of protein/mediator levels could be very interesting to reinforce the results from the mRNA quantification. However, this does not mean that our results are not appropriate or insufficient regarding the stimulation of RAW 264.7 macrophages with IL-4 or the activity of P. angulata in this context. First, we optimized the method to acquire alternative activated RAW 264.7 by testing serial concentrations of IL-4 and measuring the gene expression of ARG1 (the most important metabolic marker in mouse macrophages as well-revised by Murray, 2014 – doi:10.1016/j.immuni.2014.06.008). In this manner, we found that incubation with IL-4 (40 ng/mL) induced a significant increase in the expression of ARG1 (P<0.01, data not shown) that can also be observed in the Figure 1. As expected, IL-10 was not induced by IL-4, but we observed a trend to induce MCR1 (which is expected in human macrophages but not so frequently described in murine macrophages). Unfortunately, we have a limited amount of primers available in the lab and did not purchase fizz1, also we are not able to assay this marker to improve the paper since the shipment of primers takes at least a few weeks from the US to Colombia. Overall, we consider that stimulation of RAW264.7 with IL-4 was a successful method to obtain alternative activated macrophages. Moreover, it is frequent in the literature that only mRNA levels are used to sustain this (Zhang, 2009 – doi:10.1002/jcb.22110)
Anyway, as we routinely measured nitrite levels to confirm the metabolic status of LPS-activated macrophages (Supplementary Figure 3), we also aimed to evaluate a metabolite involved in the ARG1 pathway. We revised the literature and found out that ornithine was commonly employed and tried to purchase a kit to measure it; however, it was not possible and therefore we decided to determine urea levels. The results were satisfactory since IL-4 induced the levels of urea from 6.35±3.18 mg/dL (control, naïve macrophages) to 11.29±0.25 mg/dL (P<0.05). Taking into account your comment, we decided to include a bar showing the results for control macrophages in the Supplementary Figure 3. We also understand that it might be confusing since LPS is also raising urea levels, but it was not unexpected since LPS-activated mouse macrophages also activate the expression of ARG1 as a feedback.
Finally, we also agree with the idea of testing P. angulata in other cellular models. We did try to activate HT-29 intestinal epithelial cells with LPS (without success) and are currently optimizing activation with TNF-α. In addition, we intend to test in other human-derived cells and intestinal tissue from IBD patients. These experiments are warranted for future work.
Comment: Fig. 2: It is not clear whether the data presented in the two graphs of panel 2A are statistically significant or not. The authors should clarify it in the manuscript.
Response: There are no significant differences in the data presented in the graphs of the panel 2A, therefore it is not indicated in the figure. This was clarified in the manuscript including the sentences “with reaching statistical significance” or “also tended to” when referring to the changes observed for both parameters (Body weight and Disease Activity Index).
Comment: Fig. 3: The authors show that Physalis angulata calyces in vivo administration reduces MPO activity in intestinal samples from DSS-treated animals. While this result correlates with the results presented in Fig. 2, it is not clear how to link them with results in Fig.1. Have the authors investigated whether Physalis angulata calyces affect neutrophil gene expression and inflammatory response? Similarly, experiments on MLN do not correlate with the data on macrophages, as MLN are composed by different immune populations.
Response: We fully agree with your comment, neither the MPO activity nor the data from MLN experiments can be linked (directly) with the results obtained with RAW 264.7 macrophages. In fact, we were not able to find any sentence in the manuscript that put those results together. The in vivo model was kept separated not only in the result section but also in the discussion, and they are only related with results from macrophages when it was stated that the in vitro results are a possible explanation for the anti-inflammatory effect of the test fraction in the DSS model (“Hence, it is plausible that PADF significantly ameliorates DSS-colitis by means of macrophage transition towards an anti-inflammatory phenotype… Further research is needed to undoubtedly demonstrate whether the effect of P. angulata is mediated by intestinal macrophages instead of other cell populations”).
In addition, we do not have data regarding the activity of P. angulata on neutrophils.
Comment: Fig. 1: Physalis angulata calyces concentration in the figures is measured as mg/kg, while the legend states ug/ml.
Response: Figure 1 was revised and corrected as suggested.
Comment: Fig. 2: The authors should state the number of animals analyzed in fig. 2B and 2C, as the legend says “three independent experiments”.
Response: The number of animals was included in the legend as requested.
Reviewer 3 Report
Introduction: The introduction should provide studies about the anti-inflammatory effects of Solanaceae extracts on macrophage polarization such as https://doi.org/10.1002/jcp.27537 and all compounds with anti-inflammatory properties in its extract. Results: In Figure 1, please mention if the cells were treated with the vehicle or not. In Figure 1B, In addition to anti-inflammatory genes, the expression of pro-inflammatory genes by PDAF treatment in the absence of LPS should be assessed to be sure about its anti-inflammatory effect. Discussion: Please explain your results for IL10 in Figure 3S. Please discuss your results for Figure 3C.
Author Response
Dear Reviewer 3,
Thank you for taking the time to review our manuscript. We appreciate all your comments and suggestions and edited the document accordingly. We hope that these modifications meet your expectations. In addition, we prepared a point-by-point response to all your comments to provide further clarification.
Comment: The introduction should provide studies about the anti-inflammatory effects of Solanaceae extracts on macrophage polarization such as https://doi.org/10.1002/jcp.27537 and all compounds with anti-inflammatory properties in its extract.
Response: To our knowledge, there are around of 15 studies of the anti-inflammatory effect of extracts and compounds from P. angulata. We did not consider convenient to fully describe them in the introduction. Given your suggestion, we included the following statement “Focusing on the anti-inflammatory potential of P. angulata, numerous studies have demonstrated strong immunomodulatory function of several derivatives, with recent reports revealing the immunosuppressive effects of some withanolides due to regulation of macrophage polarization.”
Comment: In Figure 1, please mention if the cells were treated with the vehicle or not.
Response: As suggested the sentence “Naïve RAW 264.7 macrophages were treated with PADF (6.25 and 12.50 μg/mL) or vehicle (positive and negative controls)” was included in the legend.
Comment: In Figure 1B, In addition to anti-inflammatory genes, the expression of pro-inflammatory genes by PDAF treatment in the absence of unless otherwise stated LPS should be assessed to be sure about its anti-inflammatory effect.
Response: We agree with your comment, it was unwise not to include these samples during the quantification of inflammatory genes expression. However, the concentration of nitrites (as a surrogate of nitric oxide) was measured routinely and it was never observed an increase in its liberation when macrophages were treated with P. angulata.
Comment: Please explain your results for IL10 in Figure 3S.
Response: A possible explanation for these results lays in the fact that during LPS activation of macrophages, IL-10 is expressed through the induction of the nuclear factor (NF)-κB. As P. angulata reduces the liberation of NO, PGE2, IL-1β, and IL-6 (pro-inflammatory), as well as IL-10 (anti-inflammatory) after 24 h of incubation; it is reasonable to hypothesize that test fraction is blocking the activation of NF-κB.
Results presented in Figure 1B might appear confusing when comparing with those presented in Figure 3S. However, it should be kept in mind that the incubation time with LPS is different in each experimental setting (6 h in Figure 1B Vs 24 h in Figure 3S), hence the cellular status is different and these discrepancies are expected. For instance, at 6 h the mRNA expression of IL-10 in LPS-treated macrophages appears to be lower in comparison with controls (only vehicle), while its liberation in the culture media after 24 h was found to be higher (without meaning that LPS is an IL-10 inductor). In the same way, P. angulata increases IL-10 expression during the peak of the inflammatory momentum of the activated macrophage, but its liberation seems to be reduced when the cell has reached a resolutory condition.
Comment: Please discuss your results for Figure 3C.
Response: As suggested the following statement was modified “This is reinforced by the decreased production of IL-6 and MCP-1 by activated MLN cells, mediators that also play a pivotal role in the migration and attraction of inflammatory macrophages to colonic tissue when damage is induced by DSS [24].”
Round 2
Reviewer 2 Report
The authors have modified the abstract to clarify the absence of direct correlation between data on macrophages and in vivo data on a colitis mouse model. I would advice of modifying the discussion session accordingly: for example, at line 286 of the revised manuscript, instead of "it is plausible that PADF significantly ameliorates DSS-colitis by means of macrophage transition towards an anti-inflammatory phenotype" it should be stated that PADF effect on macrophages is one of the possible explanations for the data in fig. 2 and 3.
I understand that the time given for revision does not allow to perform new experiments, which could have completed and clarified the findings described in the manuscript.
Minor points:
- in the revised version, there are two figure 1.
- few minor English mistakes were present in the manuscript.